# Analysis of Business Customers' Energy Consumption Data Registered by Trading Companies in Poland

**Agnieszka Kowalska-Styczeń** [1,*], **Tomasz Owczarek** [1], **Janusz Siwy** [2], **Adam Sojda** [1] and **Maciej Wolny** [1]

1 Faculty of Organization and Management, Silesian University of Technology, 26-28 Roosevelta Street, 41-800 Zabrze, Poland; tomasz.owczarek@polsl.pl (T.O.); adam.sojda@polsl.pl (A.S.); maciej.wolny@polsl.pl (M.W.)

2 Ebicom Sp. z o.o., 65 Sokolska Street, 40-087 Katowice, Poland; janusz.siwy@ebicom.pl

* Correspondence: agnieszka.kowalska-styczen@polsl.pl

**Abstract:** In this article, we analyze the energy consumption data of business customers registered by trading companies in Poland. We focus on estimating missing data in hourly series, as forecasts of this frequency are needed to determine the volume of electricity orders on the power exchange or the contract market. Our goal is to identify an appropriate method of imputation missing data for this type of data. Trading companies expect a specific solution, so we use a procedure that allows to choose the imputation method, which will consequently improve the accuracy of forecasting energy consumption. Using this procedure, a statistical analysis of the occurrence of missing values is performed. Then, three techniques for generating missing data are selected (missing data are generated in randomly selected series without missing values). The selected imputation methods are tested and the best method is chosen based on MAE and MAPE errors.

**Keywords:** electricity consumption data; missing data imputation; effective energy management; data quality; time series





## 1. Introduction

Time series analysis is widely used in many management systems, in the areas of: transport systems, including urban transport [1–4] environment [5–8], medical data [9–15] and energy [16–21]. In this work, we analyze data on electricity consumption. As emphasized by Wang et al. [21], economic development causes an increase in electricity demand, and thus generates the need to save energy, i.e., better and better energy management systems. Such systems are mainly dedicated to electricity consumers, but effective energy management is also crucial for electricity trading companies. In their activities, in addition to commercial problems and challenges, such companies must often purchase energy on the wholesale market and then distribute it to individual customers. There is a need to ensure continuous and accurate balancing of electricity demand and production in this process. This is due to, among other things, the inability to store the purchased product, as well as the need to balance the demand for electricity with the supply at any time. Therefore, it is very valuable to know about electricity demand in the near and far horizon, i.e., energy consumption schedules. Such schedules can be defined as a set of data specifying the amount of electricity planned to be introduced or taken from the grid for particular periods (e.g., day, week, month or year). Standardization of such a schedule leads to developing a profile characteristic for a given recipient or group of recipients. Therefore, it is important in this context to increase the accuracy of forecasting electricity consumption, which depends on the quality of the collected data [7,22].

The specificity of electricity trading requires the analysis of hourly data, because forecasts of such frequency are needed to determine the volume of electricity orders on the power exchange or contract market, and then, if necessary, to correct these orders. Despite the intensive development of smart metering and the installation of an increasing number

of meters ensuring the possibility of transmitting hourly values, only a dozen or so percent (a small percentage) of concluded contracts are settled based on these measurements. In most cases, electricity distributors only provide the seller with the total amount of energy consumed by the consumer during the load period, which varies from a few days to a year. Periodic readouts are then distributed in the time series with an hourly gradation using standard profiles developed by distributors. Execution of a new contract with the customer requires preparing a consumption forecast for its whole term. To create the forecast, the data of energy consumption by the customer in preceding periods or the declared energy consumption for a new building is required. However, the acquired historical data is not often complete and contains missing values. This is a common problem when data is measured and recorded [23,24]. Various reasons lead to a lack of values in the time series. In the case of energy consumption data, these can be communication errors, sensor failures, or power outages [25], but also missing values due to the lack of readings (values are then not measured).

The extensive literature on the imputation of missing data shows algorithms for replacing missing data with estimates [26]. The most common data imputation techniques rely on correlations between attributes to estimate values for missing data. These include: Multiple Imputation [27], Expectation-Maximization [28], Nearest Neighbor [29], and Hot Deck [30]. Many studies show examples of multidimensional time series imputation [1,4,31–35]. However, in the case of univariate series, there are no additional attributes, therefore imputation algorithms specially adapted to such data should be used [23]. For example, Bokde et al. [25] propose the 'imputePSF' method, which is a modification of the pattern sequence based forecasting (PSF) method, while Demirhan and Renwick [5] compare the performance of the methods available in the 'imputeTS' package, which are dedicated to univariate time series with irregular intervals.

An important element to pay attention to when using imputation methods is the type of data. Depending on the field from which they originate, the data may be characterized by the presence of a trend, seasonality or randomness, or property known as the effect of volatility clustering (volatility in one subperiod depends on the volatility realized in preceding periods). Since series from different fields have distinct characteristics, different imputation methods give better results for the series.

In our study, we analyze anonymised data from Polish energy trading companies. These companies buy electricity wholesale and then sell it to direct customers. It should be emphasized that the trading company does not have direct access to measuring devices and does not read them. The owner of the metering devices (energy meters) from which the readings of energy consumption come is the distribution network operator (DNO). The DNO is obliged to provide the trading company with data on the consumption of the recipient for whom this company provides services related to the sale of energy. Based on these data, the trading company accounts for energy recipients and balances supply and demand on the energy market. Intensive work is underway in Poland to ensure that most of the data for billing comes from smart meters in the form of hourly readings, but at the moment, it is still a problem.

After analyzing many of the previously cited works on the imputation of missing values, we noticed a certain limitation in the applicability of the widely discussed methods and techniques to the analysis of our data. This limitation is the type of data we received from trading companies, which had the form of one-dimensional series and did not contain additional attributes (we only know energy consumption at a given point of electricity consumption—PPE). Therefore, we decided to develop a procedure adapted to the received data, which would allow us to choose the appropriate method of the imputation of the missing value, which could be used by trading companies.

Our procedure allows you to select the best of the tested imputation methods along with the error evaluation, and requires the use of a series with both missing values and no gaps. First, we perform a statistical analysis of the occurrence of missing values in the series to select the techniques for generating missing data. We then use these techniques to

generate missing data in randomly selected series without missing data. The selected methods and variants of imputation are compared based on MAPE (mean absolute percentage error) and MAE (mean absolute error) errors calculated for individual points of electricity consumption (PPE) based on real values and imputed values. A detailed description of the procedure is provided in Section 3.4. The results of our research will allow for more accurate forecasts and thus for better planning of purchases by trading companies. The presented research focuses mainly on business customers due to the frequency of readings from energy meters. For this group of customers, data are provided from smart meters similar to other European countries. The results of our work may therefore also be interesting for other energy trading markets.

## 2. Data

The quality of electricity consumption data is a critical issue in mining big data relating to the energy industry [22]. Thanks to the analysis of this data, it is possible to extract valuable knowledge to increase the level of profitability of energy companies as well as electricity trading companies. Electricity data quality issues can be divided into three categories: noise data—including logical errors and inconsistent data, incomplete data, and outlier data [22]. The problem faced by trading companies that provided data primarily concerns incomplete data, i.e., data containing missing values. As mentioned earlier, the specificity of the electricity market requires paying particular attention to hourly data and such series are discussed in this article.

Data of business customers in Poland from tariff groups B and C were analyzed in detail, as the recipients of these tariffs account for 79% of all customers of trading companies that agreed to provide data for the research. Tariff B is the Medium Voltage used by large enterprises (excluding the largest recipients such as mines or large factories), while Tariff C is the Low Voltage dedicated to small and medium-sized enterprises (mainly service and trade companies). The data of individual customers were excluded from the study, because for them, energy readings are carried out at large intervals (even every few months) and, as indicated, are not the main customers of the surveyed trading companies.

Finally, a database consisting of 3236 data series (data from 3236 PPE in 2019) was selected for the analysis with the following characteristics:

- the length of a single sequence of missing data (gaps) not longer than 48 h in one sequence,
- no more than 576 h with missing data during the year,
- no more than 20% of profile consumption (these are values estimated based on profiles prepared by trading companies in the event that the energy consumption readings occur in periods longer than every hour, e.g., once a day or once a week).

*Missing Values Analysis*

In a database of 3236 data series, 210 series (210 PPE) contained the missing values. For these series, a statistical analysis of the occurrence of missing data was performed. The number of missing values, the number of gaps (a gap is defined as one or more data missing in succession), the longest sequence of missing values, the shortest sequence of missing values, and the average length of gaps were analyzed. Details are provided in Table 1.

As shown in Table 1, only 5% of PPE has missing values in more than 58 reading positions. For the indicated points, the number of missing values is not uniformly distributed across all PPE. The number of missing observations is characterized by large right- skewed asymmetry and the presence of outliers (relatively large deviation). The distribution of the number of gaps is a consequence of the distribution of the number of missing observations and is also characterized by large right-skewed asymmetry. In at least 50% of PPEs, the number of gaps does not exceed four (see Table 1, the median for 'longest gap' is 4). Lengths of data gaps were also analyzed. As can be seen in Table 1, the longest missing substring of data was 48 (according to the criteria for selecting PPE for imputation).

Figure 1 shows the distribution of the number of missing values (without taking into account the most extreme value). The histogram of the distribution of the number of missing values confirms the earlier comments resulting from the statistical analysis of the occurrence of missing data; a large right-skewed asymmetry can be seen.

**Table 1.** Basic statistics of missing data.

| Statistics | Number of Missing Values | Number of Gaps | The Longest Sequence of Missing Values | The Shortest Sequence of Missing Values | Average Length of Gaps |
|---|---|---|---|---|---|
| min. | 1.00 | 1.00 | 1.00 | 1.00 | 1.00 |
| perc05 | 1.00 | 1.00 | 1.00 | 1.00 | 1.00 |
| perc10 | 1.00 | 1.00 | 1.00 | 1.00 | 1.00 |
| perc25 | 3.00 | 1.00 | 2.00 | 1.00 | 1.00 |
| median | 9.00 | 2.00 | 4.00 | 1.00 | 1.00 |
| perc75 | 24.00 | 5.00 | 13.75 | 2.75 | 2.75 |
| perc90 | 41.50 | 14.00 | 24.00 | 24.00 | 24.00 |
| perc95 | 58.20 | 24.55 | 24.00 | 24.00 | 24.00 |
| max. | 463.00 | 456.00 | 48.00 | 48.00 | 48.00 |
| average | 18.80 | 7.62 | 9.47 | 6.00 | 6.00 |
| std. dev. | 37.10 | 32.82 | 11.22 | 10.27 | 10.27 |
| skewness | 8.55 | 12.26 | 1.61 | 2.27 | 2.27 |

Source: own elaboration.

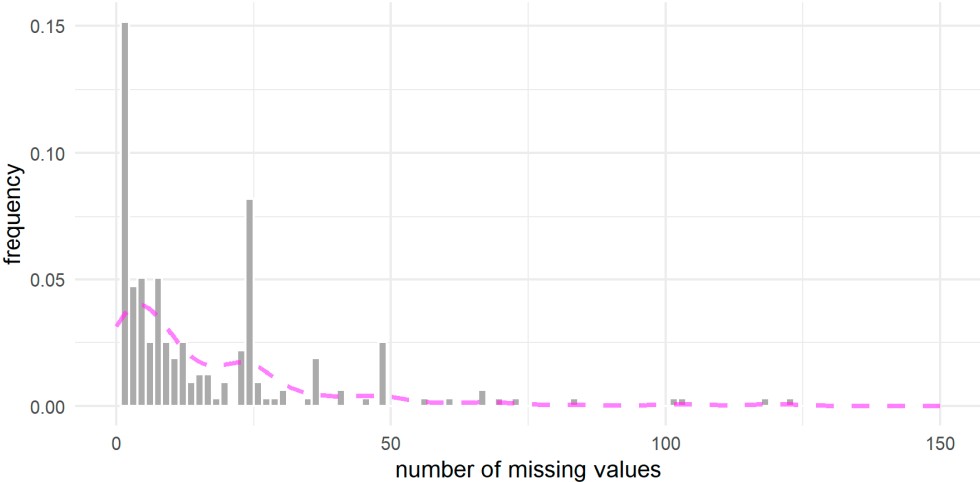

**Figure 1.** Distribution of the number of missing values. Source: own elaboration.

Then, the distribution of missing data in terms of the moment of their occurrence was examined. It was noticed that:

- most, because there were as many as 19 cases, of the missing data concerned 3:00 a.m. on 27 November 2019—the time change point (it is worth noting that on this day, the day has 25 h). However, in the second point of the time change on 31 March 2019 at 2 o'clock, no data appeared in the 14 PPE. The problems with the missing values at the time of the time change affected 29 different PPEs, but in only four cases, they occurred simultaneously in the same PPE;
- on 26 January 2019 from 1:00 to 24:00, no data appeared in 15 PPEs;
- when examining the distribution of missing values in the indicated set of 210 PPEs, it was found that out of 8760 measurement items—in 6371 (72.7%), there were no missing values in any PPE.

Examples of how missing data can be distributed in the series of consumption are shown in Figures 2–4. The figures contain hourly data from three different PPEs with missing values marked (magenta). Collection point PPE-Example1 is an example containing

103 data gaps that occur throughout the analyzed year (see Figure 2). The course of the series for which the number of missing values is 49, and they occur over a period of about one month as shown in Figure 3, while Figure 4 contains the data for the collection point for which there are 48 missing values and they form one gap of 48 h.

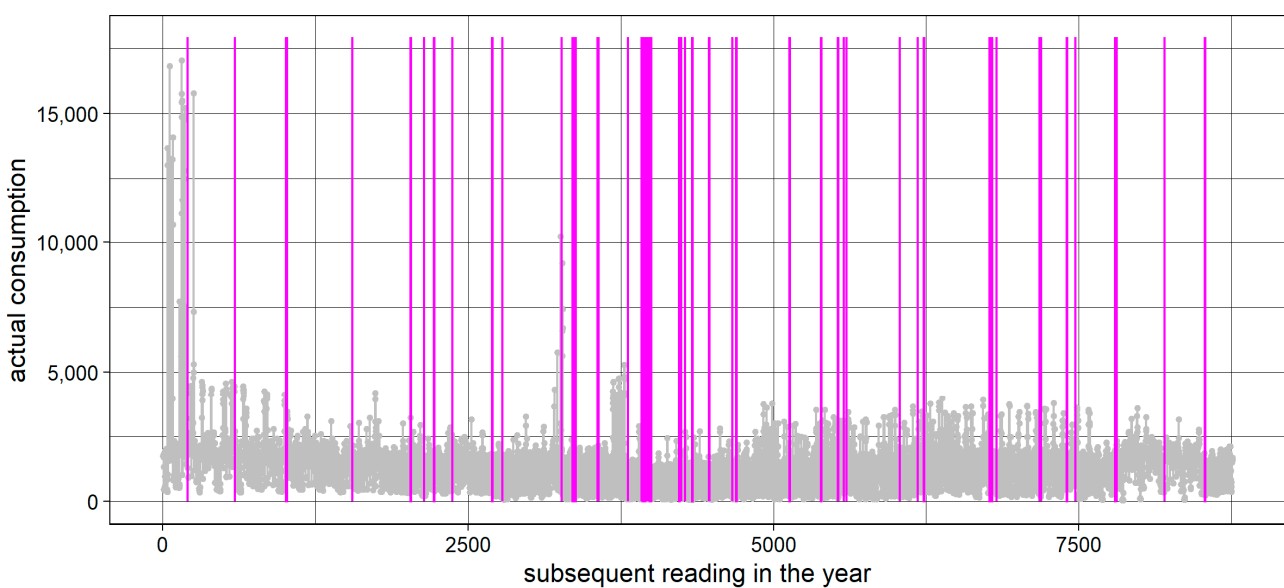

**Figure 2.** PPE-Example1—course of the series with marked missing data (103 missing values, 59 gaps, the longest sequence of missing values: 14). Source: own elaboration.

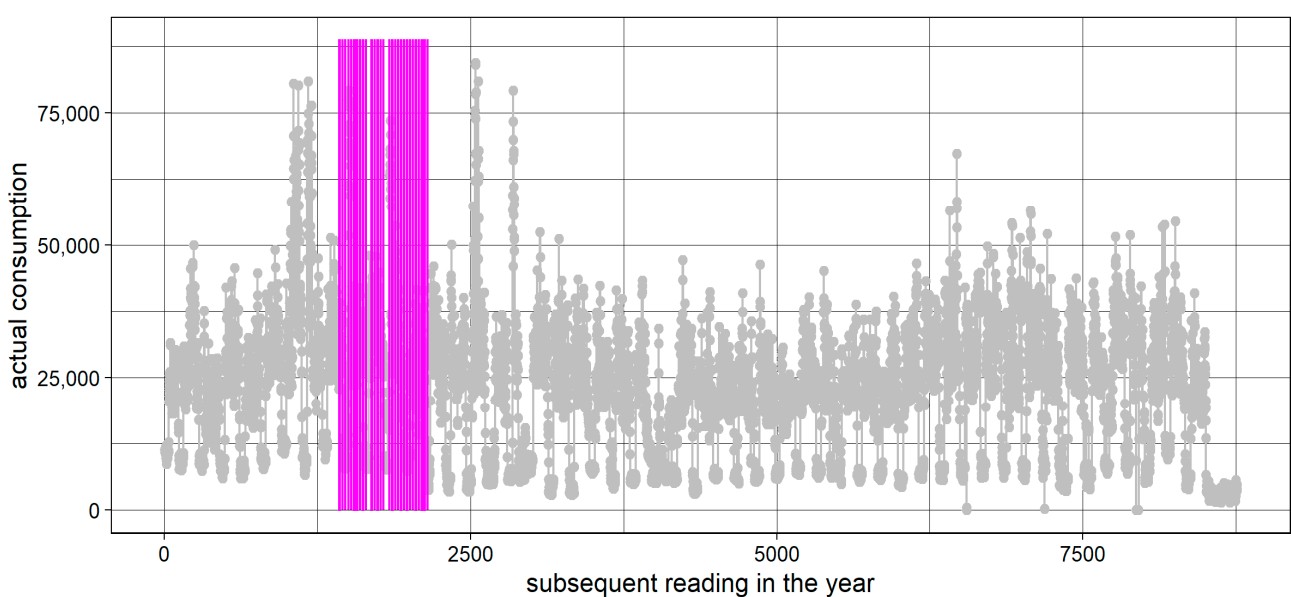

**Figure 3.** PPE-Example2—course of the series with marked missing data (48 missing values, 29 gaps, the longest sequence of missing values: 2). Source: own elaboration.

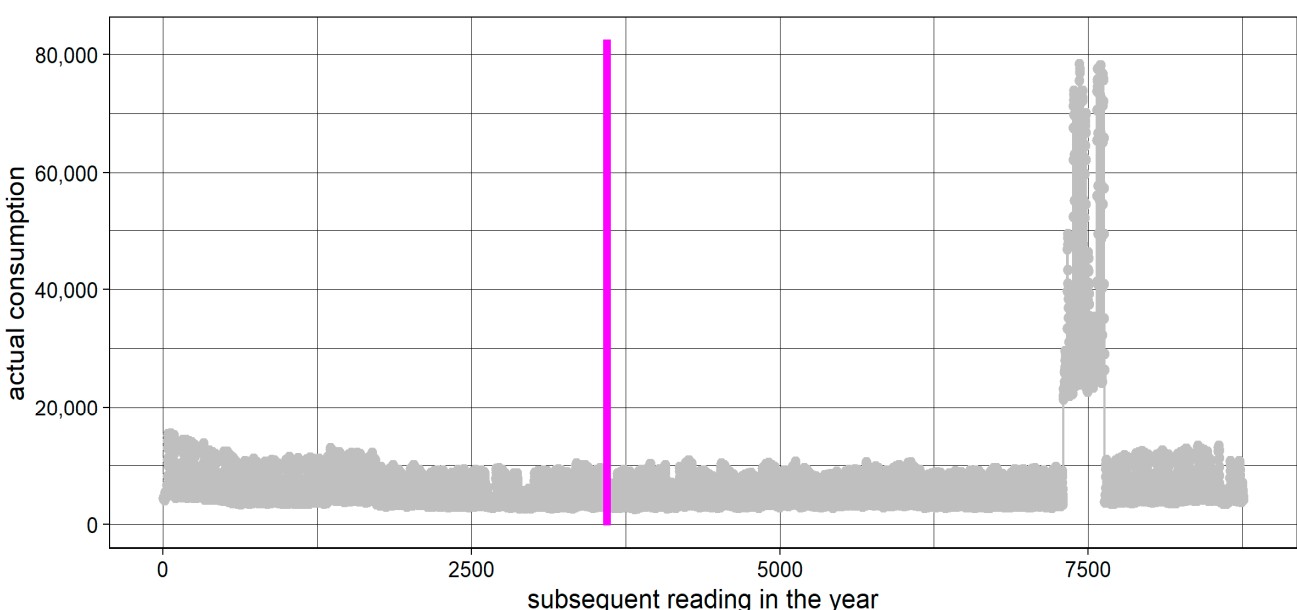

**Figure 4.** PPE-Example3—course of the series with marked missing data (48 missing values, 1 gap, the longest sequence of missing values: 48). Source: own elaboration.

When analyzing the processes of the formation of missing data, three mechanisms of their formation can be distinguished [13,24,36]:

- MCAR—missing completely at random—the process of the occurrence of missing data is considered to be completely random (there is no specific mechanism for creating missing values)
- MAR—missing at random—in the process of data occurrence, it is possible to link the occurrence of data with observable variables (there are other variables that affect the existence of missing values, and the probability of their occurrence is independent of the value itself)
- MNAR—missing not at random—in this process, missing data is related to unobservable variables (the probability of a missing value is related to the missing value itself).

It is very important to distinguish between the types of mechanisms that cause missing data because depending on the type of mechanism involved, different methods of imputation missing data will be effective to a different extent. In the case of the data provided, it can be assumed that the mechanism of missing data is of the MCAR or MAR type. Both of these mechanisms allow missing values to be assigned without knowing the specific reasons for their formation.

Taking into account the characteristics of the analyzed series and the adopted mechanism of the occurrence of MCAR or MAR missing data, three techniques for generating artificial missing data were selected:

- Random generator—single points—set (1—single). For each of the selected PPEs, 58 locations of missing data were randomly selected. The determined number of missing data was because, for 95% of PPEs with missing data, the number of missing data was not greater than 58 (see Table 1).
- Random generator—continuous data gap—set (2—continuous). For each of the selected PPEs, one gap with a length of 48 was created randomly, i.e., the longest observed missing data (see Table 1).
- Generator based on set 210—set (3—from the set). For each of the selected PPEs with complete data, one PPE was selected at random from a set of 210 imputation candidates. Missing data were inserted in the selected PPE with complete data in the places of their occurrence in the randomly selected PPE from the set of 210.

Missing data generated by the three techniques described above were used in further analyses to test imputation methods.

## 3. Methods

Many different techniques can be used to deal with missing values [13,37]. These include case deletion, mean substitution, and model-based imputation. According to Strike et al. [38], when a dataset contains less than about 10–15% of missing data, it can simply be removed from the dataset. However, it should be noted that not every dataset is subject to such rules [39], and small amounts of missing data can have a significant impact on the final result of the analysis. This is the problem we deal with in the case of given electricity consumption. As indicated earlier, the quality of such data is of great importance in increasing the accuracy of forecasting, as time series forecasts solely depend on historical data. The proper approach to dealing with missing values in the analyzed case is therefore imputation, which is one of the most reliable ways of dealing with missing values [5].

Depending on the type of data and the field from which the data comes, we have many methods of replacing missing values with estimated values. In the presented research, we have data on energy consumption, provided by energy trading companies, in the form of univariate time series. As emphasized by Moritz et al. [40], univariate time series is a particular challenge in the field of imputation research, and the time series literature focuses almost exclusively on multivariate datasets (as mentioned in the Introduction). Overall, techniques enabling imputation for univariate time series can be divided into three main categories by Moritz et al. [40]:

- One-dimensional algorithms that work with one-dimensional inputs but do not typically use time series characteristics (e.g., mean, mode, median, random sample).
- Univariate time series algorithms that can work with one-dimensional inputs but use time series characteristics. These are algorithms such as last observation carried forward, next observation carried forward, arithmetic smoothing and linear interpolation, and more advanced methods based on structured time series models that deal with seasonality.
- Multivariate algorithms on lagged data, which generally cannot be used for univariate series, but it is possible to add time information as covariates, which allows the use of multivariate imputation algorithms. This can usually be done by using lags (which take the value of another variable from the preceding period) and 'leads' (which take the value of another variable in the next period).

In this article, we are looking for solutions that will make the imputation task simple for practitioners (in our case, for trading companies). Therefore, we tested imputation methods that are dedicated to univariate data series, and for testing, we used the R package called 'imputeTS' [23].

As mentioned earlier, the subject of the study is the time series of electricity consumption in enterprises. These series are characterized by seasonality, which is related to the cyclical nature of the work of enterprises. The seasonality of the analyzed series was confirmed using the 'seastests' package in R. From the set of 3026 series with complete data, 500 series were selected at random. The conducted tests showed that all series were characterized by seasonality.

Analyses based on the characteristics of the tested time series (PPE consumption) prompted us to finally choose three methods of the imputation of missing data: the calendar method, the imputation method by separating the phases of seasonal cycles and the imputation method using seasonality decomposition. Each of the methods was used in three variants related to the seasonality of the time series and the method of taking into account the information used for imputation. Variants of each method relied on the use of a moving average with different ways of incorporating the information 'closest' to the time of the missing values.

### 3.1. Method 1—Calendar Method and 2k Weighted Moving Average Method

The main assumption of this method is taking into account the calendar and dividing the year into subseries. Each subseries refers to a specific time on a specific working day of the week or a specific time of a non-working day (the so-called 'red' days). Thus, a single subseries is, e.g., 1 p.m. on working Mondays or 5 p.m. on non-working days.

The moving average algorithm implemented in the 'imputeTS' package of the R program was used to impute the missing values. This package is recommended for imputing missing values in time series. The algorithm imputes the missing data with the mean value of the $k$ nearest values 'before' and 'after' the missing values in the series ($2k$ values in total). It was decided that the information necessary for imputation should cover a period of approximately one or two months, therefore $k = 2$ or $k = 4$ was adopted. The analyses conducted for other values of this parameter confirmed the validity of the findings.

All available methods of weight determination were used in the conducted analyses. The moving average method was with *Exponential Weighted Moving Average*, *Linear Weighted Moving Average* and *Simple Moving Average*.

Exponential weights use the information ageing principle and decrease exponentially with the distance from the missing values—'observations directly next to the central value $i$, have a weight of $\frac{1}{2}$, the observations one further away ($i − 2$, $i + 2$) have a weight of $\left(\frac{1}{2}\right)^2$ etc.'. The value of $i$ denotes the number where there is the missing value in the series. Standardized values of exponential weights are determined according to the following Formula (1):

$$w_{j,exponential} = 2^{-(j+1)} \left( \sum_{j=1}^{k} 2^{-j} \right)^{-1} \tag{1}$$

where $j$ is the distance from the missing value (in the immediate vicinity $j = 1$). For $k = 2$, $w_{1,exponential} = 0.333$, $w_{2,exponential} = 0.167$. For $k = 4$, $w_{1,exponential} = 0.267$, $w_{2,exponential} = 0.133$, $w_{3,exponential} = 0.067$, $w_{4,exponential} = 0.033$.

Linear weighted moving averages also use the information ageing principle, with the denominators of non-standard weights increasing arithmetically—'the observations directly next to a central value, have weight $1/2$, the observations one further away ($i − 2$, $I + 2$) have a weight $1/3$, etc.'. Therefore, the following weights have the following values: $\frac{1}{2}$, $\frac{1}{3}$, $\frac{1}{4}$ … The values of the standardised weights (the number of weights is $2k$) can be determined according to the Formula (2):

$$w_{j,linear} = \left[ 2(j+1) \sum_{j=1}^{k} (j+1)^{-1} \right]^{-1} \tag{2}$$

For $k = 2$, $w_{1,linear} = 0.300$, $w_{2,linear} = 0.200$. For $k = 4$, $w_{1,linear} = 0.195$, $w_{2,linear} = 0.130$, $w_{3,linear} = 0.097$, $w_{4,linear} = 0.078$. In the case of a simple moving average, the weights are the same for each value: $w_{j,simple} = [2 \cdot k]^{-1}$. For $k = 2$, $w_{1,simple} = 0.250$, $w_{2,simple} = 0.250$.

For correct operation, the algorithm requires at least two real observations to impute a missing value. In the analyzed series, a special series was the series in which the missing value was 7200th hour of the year (25th hour on the day of the time change). In the event of a missing value at that hour, imputation was performed using the moving average algorithm for the entire series. Therefore, in this special situation, the first two values from the hours before the missing value and the first two values after the missing value were taken into account (always $k = 2$).

### 3.2. Method 2—Imputation Using Seasonally Splitted Missing Value Imputation

This method relies on imputation by splitting the phases of the seasonal cycles and is implemented in the 'ImputeTS' package as the '*na_seasplit()*' function. Its idea is to split the time series into subseries defined by the phases (seasons) of the seasonal fluctuation cycles,

and then impute the values based on the separated seasons. In the conducted analyses, the moving average algorithm was used with the same parameters as in the case of the calendar method. After preliminary analyses, the number of phases was estimated at 168 (1 week = 24 h × 7 days). Therefore, the application of this method consists in distinguishing 168 subseries related to a specific time of the week, and imputation of missing values on the appropriate subseries and the weighted moving average method ($k = 2$ or $k = 4$). Additionally, in this method, all three available weighting methods were considered.

### 3.3. Method 3—Imputation Using Seasonally Decomposed Missing Value Imputation

The third method used relied on decomposing the seasonal component from the time series (in the form of a seasonality index), making imputations of missing data on the series without a seasonal component, and then reconsidering the seasonal component. This method is also implemented in the 'imputeTS' package as a *'na_seadec()'* function. Additionally, in this case, the number of phases was considered to be 168 and the weighted moving average algorithm with the values $k = 2$ or $k = 4$ was adopted. As shown in the preliminary analyses, a special feature of this method is the ability to generate values outside the acceptable range of variation (negative values). This is due to the correction of the imputed value with the value of the seasonality index. Therefore, in the conducted analyses, a correction to the implemented method was taken into account. The correction consisted in the fact that when the algorithm generated a negative imputed value, this value was changed to zero.

### 3.4. Comparison of Selected Imputation Methods

The energy consumption data analyzed in this paper contain missing data in actual values. The performance of the imputation methods used was therefore checked for simulated missing data. The procedure for selecting the appropriate missing value imputation method is shown in Figure 5 and can be described step by step as follows:

- Step 1. Select from database PPE with missing data.
- Step 2. Perform an analysis of missing data. Determine the number and distribution of missing values.
- Step 3. Prepare techniques for generating missing data adequate to the results of the analysis.
- Step 4. Select a random PPE group from the PPE database without missing data.
- Step 5. Generate missing data according to the generation techniques prepared in step 3.
- Step 6. Apply the selected imputation methods on the series from step 4.
- Step 7. Determine the accuracy of imputation methods based on MAE and MAPE errors.
- Step 8. Select the data imputation method.

To sum up, the acquired database contained 3236 time series. Each of them came from one of several energy suppliers for PPE. In the analyzed database, 210 series contained missing values that had to be imputed. The time series of electricity consumption concerned the B and C business tariffs. We did not have additional information about PPE, such as geographic location or type of business activity. The analysis of the occurrence of missing values in the series allowed for the determination of techniques for generating the missing data. These techniques were used to generate missing values in 500 randomly selected time series. We had information about the actual values in the locations of missing data, and based on this, it was possible to evaluate the indicated imputation methods.

The experiments were carried out for three imputation methods, each method was used in three variants (the moving average method with *Exponential Weighted Moving Average, Linear Weighted Moving Average* and *Simple Moving Average*) and each variant was tested for $k = 2$ for $k = 4$. This gave us 18 test cases for each technique of generating missing data.

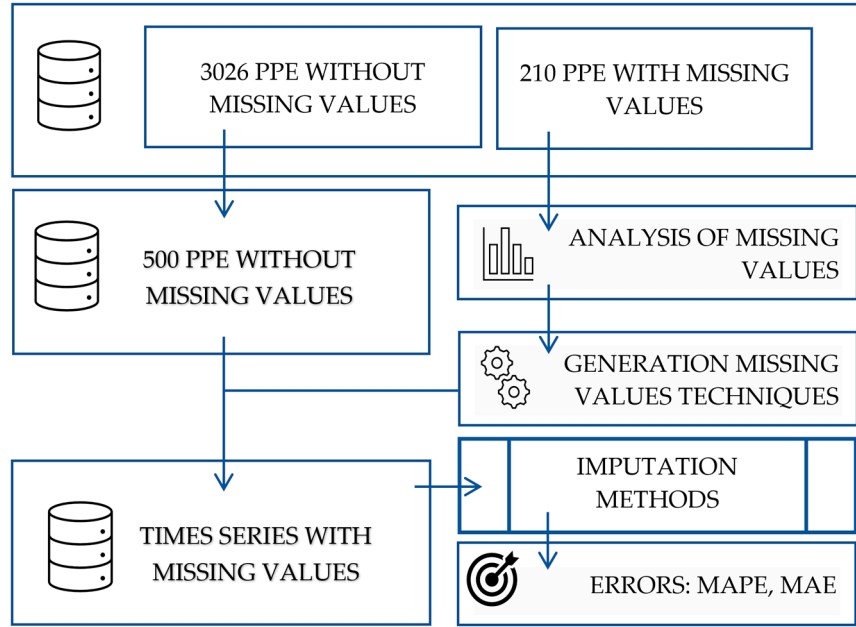

**Figure 5.** The procedure for selecting the method of the imputation of missing data. Source: own elaboration.

The selected methods and variants of imputation were compared based on the *MAPE* mean absolute percentage error and the *MAE* mean absolute error calculated for each PPE based on the actual values and imputed values. These are commonly used metrics to evaluate the performance of imputation methods for time series [21,41,42]. *MAE* measures the mean size of the errors in the forecast set without taking into account their direction, and *MAPE* is used to express the mean difference of the absolute errors between the actual and the forecasted values as a percentage of the actual values.

The error *MAPE* was determined according to the Formula (3):

$$MAPE = \frac{1}{n_{Imp_0}} \sum_{i \in Imp_0} \frac{|R_i - I_i|}{R_i} \tag{3}$$

where:

$Imp_0$—a set of indexes of readings for which data has been imputed, with no values for which $R_i = 0$,
$n_{Imp_0}$—number of inserted missing values with no values for which $R_i = 0$,
$R_i$—value of actual consumption for the generated missing data,
$I_i$—imputed consumption value.

The size shows by how much on average the imputed values differed from the actual values for a given PPE, e.g., a value of 0.05 means that the imputed values differed on average by 5%.

The error *MAE* was determined according to the Formula (4):

$$MAE = \frac{1}{n_{Imp}} \sum_{i \in Imp} |R_i - I_i| \tag{4}$$

where:

$Imp$—a set of indexes of readings for which data has been imputed,
$n_{Imp}$—the number of inserted missing values,
$R_i$—value of actual consumption for the generated missing data,
$I_i$—imputed consumption value.

The size informs by how much on average the imputed values differed from the actual values for a given PPE, e.g., a value of 500 means that the imputed values differed on average by 500.

## 4. Results and Discussion

As mentioned earlier, the variants of the imputation methods were compared based on *MAPE* (mean absolute percentage error) and *MAE* (mean absolute error) errors. Error statistics are presented for each variant of the adopted method divided into three imputation methods. Designations of variants of the tested imputation methods were constructed as follows: *method_weights_period*, e.g., Notation *2_linear_4* means method 2 (imputation with phase/season split) with linear weights and $k = 4$ (4 closest values 'before' and 4 nearest values 'after' missing)

Figure 6 shows the boxplots of MAPE values for individual methods and their variants broken down into 3 methods of generating missing values. Extremely high values (outliers) were removed from the plot for greater clarity.

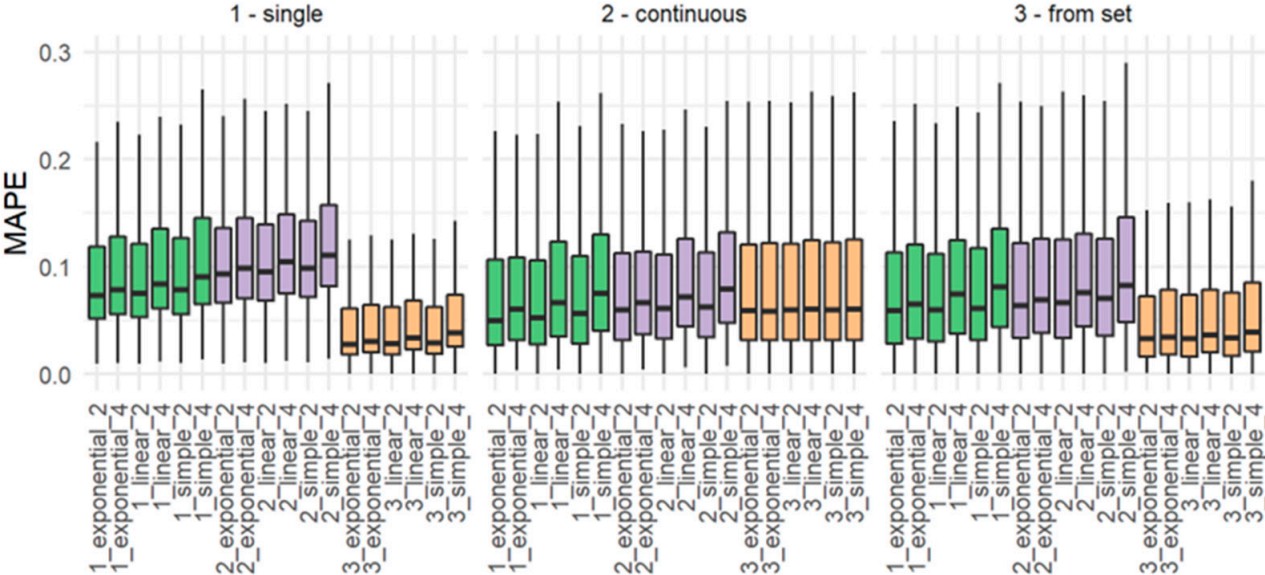

**Figure 6.** Quartiles of MAPE values for individual imputation methods. Source: own elaboration.

Figure 6 shows that in the case of the first set (1—single) and the third set (3—from the set) of generated missing data, imputation method 3 is the most effective. The quartile values for the MAPE error are clearly lower when it is used than for the other methods. Moreover, the best variant of this method is the exponential weights and $k = 2$. For the second set with missing data, the results are not so unambiguous (method 3 retains the greatest stability for its various variants, but method 1 with the value of $k = 2$ gives a lower error value).

Figure 7 shows the average values and the 95th percentile values of the MAPE error for individual variants. Similar to earlier, in the case of the first and third method of generating missing values, we can see greater efficiency of the imputation method 3. For the second method of generating missing values (middle panel), there is a clear advantage of imputation method 3 in terms of the average error value.

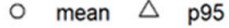

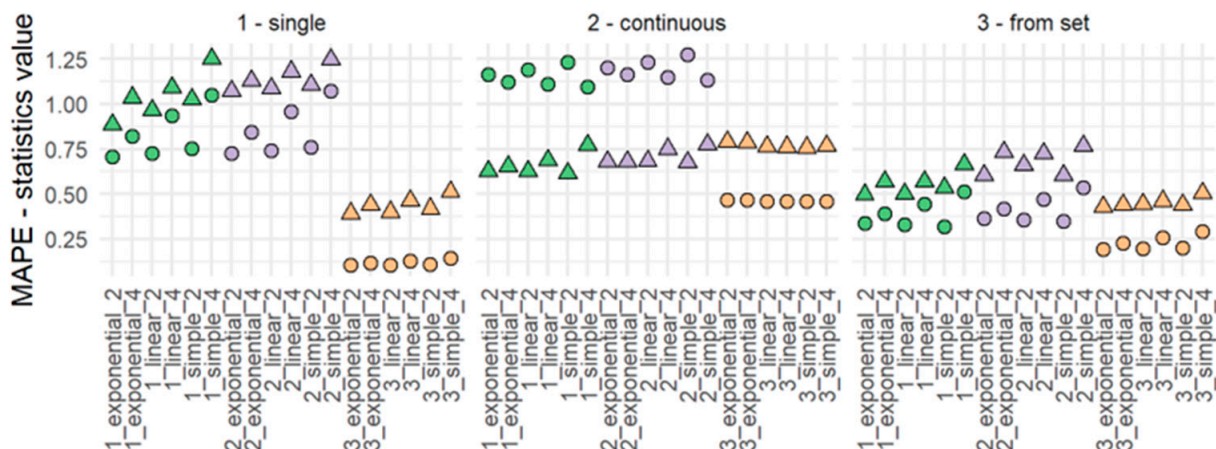

**Figure 7.** MAPE—mean and quantile 95 for sets and methods. Source: own elaboration.

Figures 8 and 9 present the distributions of MAPE and MAE errors for one variant (exponential weights and $k = 2$) of each of the three imputation methods. A limited range of *X*-axis values was presented because extremely high values (mainly for methods 1 and 2) disturbed the readability of the figures. It can be read from both figures that for the missing values sets 1—random and 3—from the set, the third method of imputation (the lowest row of panels in Figures 8 and 9) has error values more concentrated around zero than the other two methods. This confirms the previous results and proves lower average imputation errors for this method. For the second missing data generation method (2—continuous), the results are similar for all three imputation methods.

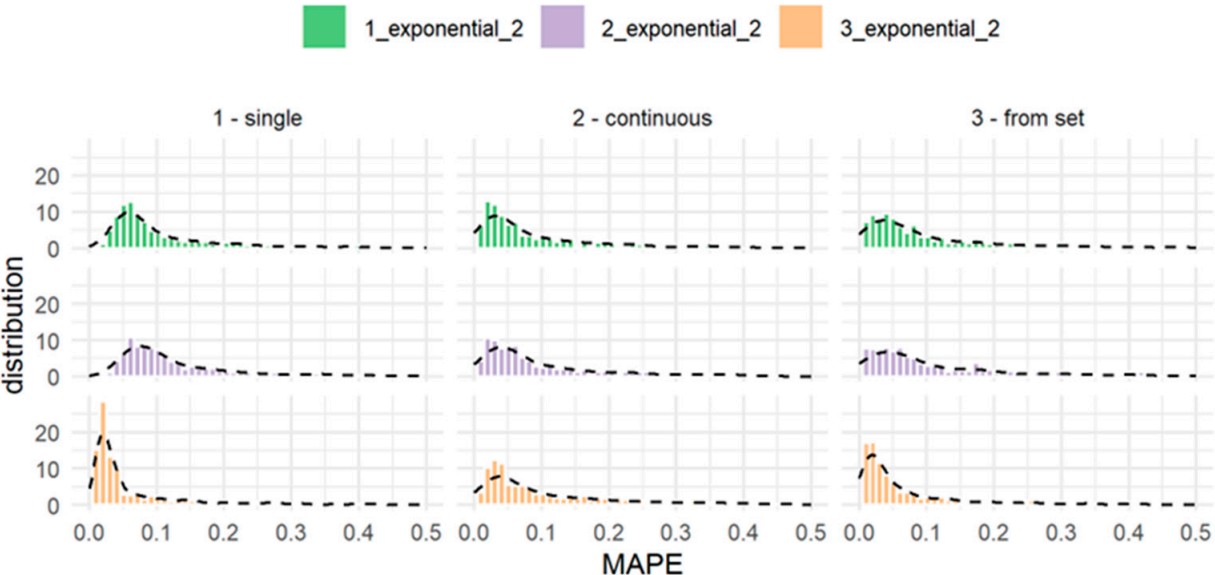

**Figure 8.** MAPE error distribution. Source: own elaboration.

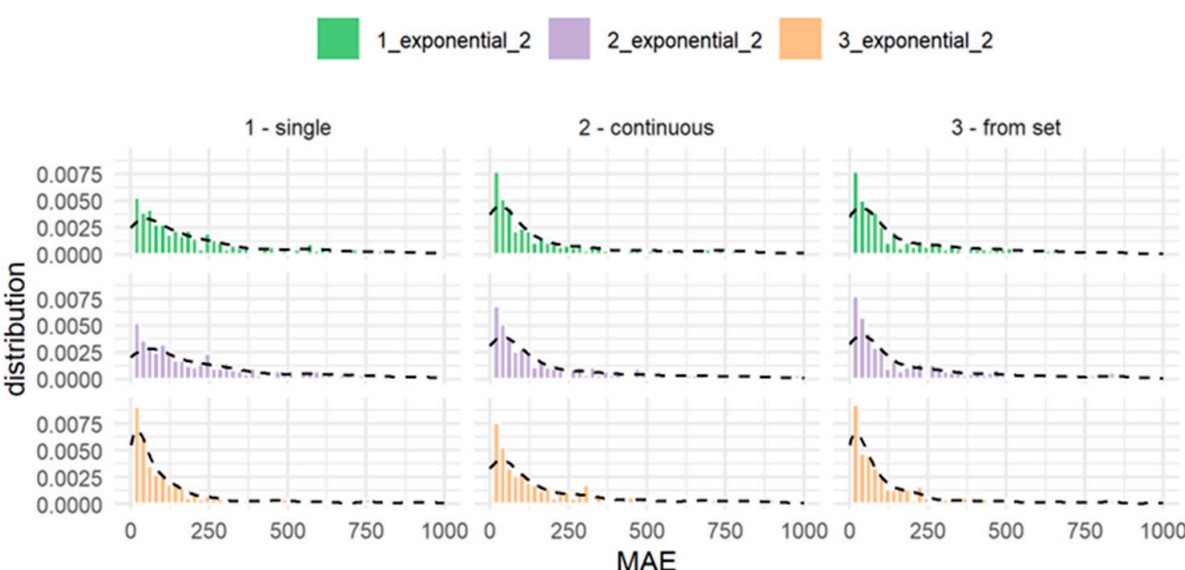

**Figure 9.** MAE error distribution. Source: own elaboration.

Tables 2–7 show detailed statistics for errors MAPE and MAE for the three methods of generating missing values.

**Table 2.** MAPE statistics for the first set of missing data (1—single).

| Type | Median | Average | Std. dev. | Q1 | Q3 | P95 | Maximum |
|---|---|---|---|---|---|---|---|
| 1_exponential_2 | 0.0839 | 0.7079 | 6.7368 | 0.0558 | 0.1767 | 0.8863 | 138.7501 |
| 1_exponential_4 | 0.0928 | 0.8223 | 7.4717 | 0.0592 | 0.1851 | 1.0359 | 135.6493 |
| 1_linear_2 | 0.0862 | 0.7250 | 6.6871 | 0.0576 | 0.1804 | 0.9667 | 133.3940 |
| 1_linear_4 | 0.1015 | 0.9349 | 8.4721 | 0.0643 | 0.2132 | 1.0910 | 131.2762 |
| 1_simple_2 | 0.0897 | 0.7515 | 6.6847 | 0.0600 | 0.1824 | 1.0264 | 125.3621 |
| 1_simple_4 | 0.1112 | 1.0507 | 9.7079 | 0.0711 | 0.2287 | 1.2522 | 167.9605 |
| 2_exponential_2 | 0.1064 | 0.7271 | 6.3287 | 0.0726 | 0.2012 | 1.0732 | 128.1718 |
| 2_exponential_4 | 0.1119 | 0.8446 | 7.1924 | 0.0762 | 0.2221 | 1.1276 | 127.3939 |
| 2_linear_2 | 0.1086 | 0.7397 | 6.2139 | 0.0739 | 0.2066 | 1.0861 | 120.6060 |
| 2_linear_4 | 0.1225 | 0.9561 | 8.2146 | 0.0825 | 0.2371 | 1.1803 | 128.4239 |
| 2_simple_2 | 0.1126 | 0.7592 | 6.1360 | 0.0769 | 0.2128 | 1.1063 | 109.2599 |
| 2_simple_4 | 0.1315 | 1.0714 | 9.4580 | 0.0890 | 0.2558 | 1.2476 | 166.3510 |
| 3_exponential_2 | 0.0304 | 0.1039 | 0.3207 | 0.0185 | 0.0784 | 0.3926 | 5.3415 |
| 3_exponential_4 | 0.0328 | 0.1151 | 0.3604 | 0.0209 | 0.0805 | 0.4403 | 5.7078 |
| 3_linear_2 | 0.0311 | 0.1065 | 0.3305 | 0.0192 | 0.0791 | 0.4002 | 5.5401 |
| 3_linear_4 | 0.0368 | 0.1282 | 0.4067 | 0.0234 | 0.0903 | 0.4614 | 6.1266 |
| 3_simple_2 | 0.0323 | 0.1106 | 0.3457 | 0.0202 | 0.0820 | 0.4189 | 5.8393 |
| 3_simple_4 | 0.0411 | 0.1422 | 0.4548 | 0.0264 | 0.0951 | 0.5120 | 6.5195 |

Source: own elaboration.

**Table 3.** MAPE statistics for the second set of missing data (2—continuous).

| Type | Median | Average | Std. dev. | Q1 | Q3 | P95 | Maximum |
|---|---|---|---|---|---|---|---|
| 1_exponential_2 | 0.0610 | 1.1626 | 19.8305 | 0.0289 | 0.1462 | 0.6266 | 441.9988 |
| 1_exponential_4 | 0.0724 | 1.1232 | 18.0315 | 0.0349 | 0.1615 | 0.6530 | 400.9493 |
| 1_linear_2 | 0.0630 | 1.1907 | 20.2304 | 0.0304 | 0.1588 | 0.6275 | 450.7924 |
| 1_linear_4 | 0.0827 | 1.1089 | 16.9140 | 0.0396 | 0.1707 | 0.6889 | 374.9546 |
| 1_simple_2 | 0.0677 | 1.2332 | 20.8335 | 0.0311 | 0.1589 | 0.6162 | 463.9826 |
| 1_simple_4 | 0.0926 | 1.0946 | 15.8026 | 0.0472 | 0.1877 | 0.7717 | 348.6892 |
| 2_exponential_2 | 0.0687 | 1.2020 | 20.0952 | 0.0355 | 0.1687 | 0.6805 | 447.9329 |
| 2_exponential_4 | 0.0785 | 1.1615 | 18.2931 | 0.0432 | 0.1861 | 0.6788 | 406.8329 |
| 2_linear_2 | 0.0727 | 1.2307 | 20.5044 | 0.0384 | 0.1729 | 0.6833 | 456.9361 |
| 2_linear_4 | 0.0906 | 1.1467 | 17.1765 | 0.0502 | 0.1971 | 0.7504 | 380.8784 |
| 2_simple_2 | 0.0770 | 1.2742 | 21.1213 | 0.0413 | 0.1848 | 0.6754 | 470.4410 |
| 2_simple_4 | 0.1046 | 1.1321 | 16.0648 | 0.0541 | 0.2183 | 0.7750 | 354.6354 |
| 3_exponential_2 | 0.0748 | 0.4681 | 5.2281 | 0.0358 | 0.1891 | 0.7914 | 115.6622 |
| 3_exponential_4 | 0.0759 | 0.4679 | 5.2271 | 0.0360 | 0.1897 | 0.7871 | 115.6617 |
| 3_linear_2 | 0.0759 | 0.4589 | 5.1854 | 0.0356 | 0.1857 | 0.7638 | 115.0321 |
| 3_linear_4 | 0.0759 | 0.4589 | 5.1840 | 0.0355 | 0.1872 | 0.7619 | 115.0314 |
| 3_simple_2 | 0.0759 | 0.4579 | 5.1803 | 0.0357 | 0.1860 | 0.7581 | 114.9646 |
| 3_simple_4 | 0.0760 | 0.4583 | 5.1793 | 0.0353 | 0.1884 | 0.7667 | 114.9636 |

Source: own elaboration

**Table 4.** MAPE statistics for the third set of missing data (3—from the set).

| Type | Median | Average | Std. dev. | Q1 | Q3 | P95 | Maximum |
|---|---|---|---|---|---|---|---|
| 1_exponential_2 | 0.0698 | 0.3369 | 3.1054 | 0.0341 | 0.1631 | 0.4969 | 67.0417 |
| 1_exponential_4 | 0.0761 | 0.3900 | 3.0152 | 0.0387 | 0.1736 | 0.5692 | 57.1944 |
| 1_linear_2 | 0.0711 | 0.3297 | 2.8744 | 0.0361 | 0.1623 | 0.5008 | 61.5542 |
| 1_linear_4 | 0.0856 | 0.4432 | 3.5166 | 0.0445 | 0.1936 | 0.5701 | 60.6301 |
| 1_simple_2 | 0.0771 | 0.3194 | 2.5339 | 0.0379 | 0.1623 | 0.5344 | 53.3229 |
| 1_simple_4 | 0.0974 | 0.5117 | 4.6669 | 0.0493 | 0.2088 | 0.6651 | 95.9030 |
| 2_exponential_2 | 0.0762 | 0.3644 | 3.0108 | 0.0380 | 0.1749 | 0.6030 | 63.3403 |
| 2_exponential_4 | 0.0847 | 0.4182 | 2.9820 | 0.0438 | 0.1851 | 0.7336 | 54.7014 |
| 2_linear_2 | 0.0795 | 0.3577 | 2.7827 | 0.0396 | 0.1762 | 0.6618 | 57.9167 |
| 2_linear_4 | 0.0925 | 0.4692 | 3.4848 | 0.0501 | 0.2008 | 0.7265 | 60.6301 |
| 2_simple_2 | 0.0810 | 0.3481 | 2.4475 | 0.0412 | 0.1768 | 0.6051 | 49.7812 |
| 2_simple_4 | 0.1022 | 0.5352 | 4.6400 | 0.0562 | 0.2197 | 0.7672 | 95.9030 |
| 3_exponential_2 | 0.0359 | 0.1926 | 1.3696 | 0.0173 | 0.0965 | 0.4283 | 28.9840 |
| 3_exponential_4 | 0.0381 | 0.2257 | 1.5628 | 0.0192 | 0.1041 | 0.4395 | 27.2371 |
| 3_linear_2 | 0.0370 | 0.1960 | 1.3779 | 0.0175 | 0.1011 | 0.4446 | 29.1219 |
| 3_linear_4 | 0.0394 | 0.2587 | 1.9901 | 0.0216 | 0.1127 | 0.4611 | 34.6183 |
| 3_simple_2 | 0.0371 | 0.1997 | 1.3880 | 0.0175 | 0.1021 | 0.4404 | 29.3057 |
| 3_simple_4 | 0.0441 | 0.2924 | 2.5204 | 0.0229 | 0.1200 | 0.5053 | 49.2378 |

Source: own elaboration.

**Table 5.** MAE statistics for the first set of missing data (1—single).

| Type | Median | Average | Std. dev. | Q1 | Q3 | P95 | Maximum |
|---|---|---|---|---|---|---|---|
| 1_exponential_2 | 192.1606 | 1435.9731 | 4922.151 | 6716.119 | 666.1800 | 6716.119 | 49,863.34 |
| 1_exponential_4 | 207.4420 | 1467.3281 | 4985.422 | 6669.177 | 701.3838 | 6669.177 | 48,732.12 |
| 1_linear_2 | 196.0299 | 1459.5901 | 5021.913 | 6540.056 | 686.6782 | 6540.056 | 50,309.75 |
| 1_linear_4 | 227.2977 | 1533.7125 | 5178.168 | 7153.797 | 734.2430 | 7153.797 | 51,900.50 |
| 1_simple_2 | 206.0445 | 1502.3407 | 5194.361 | 6641.748 | 714.4662 | 6641.748 | 51,531.03 |
| 1_simple_4 | 255.2485 | 1619.9814 | 5453.083 | 7181.493 | 812.5138 | 7181.493 | 55,188.93 |
| 2_exponential_2 | 244.8197 | 1606.4145 | 5679.998 | 6803.326 | 755.4154 | 6803.326 | 65,369.56 |
| 2_exponential_4 | 253.4706 | 1621.4475 | 5635.571 | 6836.370 | 784.0457 | 6836.370 | 64,811.75 |
| 2_linear_2 | 247.9669 | 1629.7881 | 5779.796 | 6632.659 | 766.1345 | 6632.659 | 67,139.35 |
| 2_linear_4 | 273.5827 | 1685.8397 | 5844.150 | 7064.499 | 830.4936 | 7064.499 | 69,261.74 |
| 2_simple_2 | 254.5287 | 1672.3070 | 5953.187 | 6776.913 | 785.7015 | 6776.913 | 69,962.12 |
| 2_simple_4 | 297.1194 | 1777.5822 | 6181.560 | 7149.959 | 887.0493 | 7149.959 | 75,569.45 |
| 3_exponential_2 | 80.0103 | 667.3803 | 2167.906 | 2906.804 | 332.5147 | 2906.804 | 26,231.43 |
| 3_exponential_4 | 85.5530 | 687.7242 | 2218.258 | 3009.515 | 352.9507 | 3009.515 | 25,449.99 |
| 3_linear_2 | 81.1205 | 676.9495 | 2193.493 | 2923.097 | 341.9984 | 2923.097 | 26,307.31 |
| 3_linear_4 | 93.9804 | 726.0771 | 2328.082 | 3090.166 | 383.9144 | 3090.166 | 25,450.96 |
| 3_simple_2 | 83.1870 | 694.8155 | 2245.569 | 2945.131 | 355.6474 | 2945.131 | 26,606.25 |
| 3_simple_4 | 100.0570 | 778.6808 | 2509.006 | 3271.714 | 401.5057 | 3271.714 | 26,981.26 |

Source: own elaboration.

**Table 6.** MAE statistics for the second set of missing data (2—continuous).

| Type | Median | Average | Std. dev. | Q1 | Q3 | P95 | Maximum |
|---|---|---|---|---|---|---|---|
| 1_exponential_2 | 125.5122 | 1303.307 | 5915.684 | 6045.936 | 672.8637 | 6045.936 | 110,622.9 |
| 1_exponential_4 | 147.0003 | 1338.060 | 6168.844 | 6077.178 | 701.9186 | 6077.178 | 119,203.2 |
| 1_linear_2 | 124.7875 | 1305.832 | 5882.215 | 6098.225 | 675.9620 | 6098.225 | 110,724.1 |
| 1_linear_4 | 173.0705 | 1389.446 | 6405.333 | 6133.712 | 770.1902 | 6133.712 | 125,824.5 |
| 1_simple_2 | 135.5104 | 1316.439 | 5859.758 | 5914.193 | 690.8815 | 5914.193 | 111,086.7 |
| 1_simple_4 | 204.6888 | 1460.388 | 6706.472 | 6292.746 | 833.4499 | 6292.746 | 132,736.3 |
| 2_exponential_2 | 140.1250 | 1552.421 | 8579.709 | 6564.128 | 673.7960 | 6564.128 | 176,030.7 |
| 2_exponential_4 | 169.1431 | 1535.459 | 7920.725 | 6186.559 | 719.6874 | 6186.559 | 160,775.8 |
| 2_linear_2 | 150.9187 | 1537.193 | 8223.784 | 6471.902 | 696.2083 | 6471.902 | 167,956.6 |
| 2_linear_4 | 197.9992 | 1539.498 | 7264.910 | 6428.482 | 780.9942 | 6428.482 | 144,291.2 |
| 2_simple_2 | 166.2839 | 1520.573 | 7706.642 | 6316.463 | 737.4375 | 6316.463 | 155,845.3 |
| 2_simple_4 | 220.8806 | 1570.909 | 6783.569 | 6371.571 | 866.1808 | 6371.571 | 130,590.6 |
| 3_exponential_2 | 149.9771 | 1545.683 | 8837.987 | 6613.334 | 747.6411 | 6613.334 | 183,595.5 |
| 3_exponential_4 | 149.9898 | 1544.193 | 8834.488 | 6608.279 | 754.0734 | 6608.279 | 183,655.0 |
| 3_linear_2 | 149.8648 | 1546.627 | 9160.846 | 6533.728 | 744.0286 | 6533.728 | 192,179.4 |
| 3_linear_4 | 149.8723 | 1546.353 | 9159.699 | 6515.926 | 746.2735 | 6515.926 | 192,288.8 |
| 3_simple_2 | 149.6098 | 1546.969 | 9193.438 | 6556.608 | 737.3952 | 6556.608 | 193,132.0 |
| 3_simple_4 | 150.1604 | 1549.090 | 9195.696 | 6553.450 | 745.6235 | 6553.450 | 193,282.1 |

Source: own elaboration.

**Table 7.** MAE statistics for the third set of missing data (3—from the set).

| Type | Median | Average | Std. dev. | Q1 | Q3 | P95 | Maximum |
|---|---|---|---|---|---|---|---|
| 1_exponential_2 | 134.9881 | 1814.0505 | 10,237.756 | 6070.180 | 615.6667 | 6070.180 | 197,800.00 |
| 1_exponential_4 | 153.0144 | 1851.1807 | 9985.260 | 6942.875 | 612.4934 | 6942.875 | 190,863.00 |
| 1_linear_2 | 135.6714 | 1827.7775 | 10,229.210 | 6479.405 | 626.6606 | 6479.405 | 197,200.00 |
| 1_linear_4 | 176.5047 | 1895.5239 | 9574.763 | 7516.181 | 705.8925 | 7516.181 | 178,201.82 |
| 1_simple_2 | 141.3125 | 1852.7049 | 10,232.072 | 6688.394 | 654.7261 | 6688.394 | 196,300.00 |
| 1_simple_4 | 197.4844 | 1967.2077 | 9241.462 | 8426.872 | 763.3424 | 8426.872 | 164,636.25 |
| 2_exponential_2 | 143.8694 | 2007.7018 | 10,737.351 | 7074.631 | 680.7083 | 7074.631 | 197,800.00 |
| 2_exponential_4 | 159.3135 | 2040.3169 | 10,506.338 | 7270.989 | 751.0931 | 7270.989 | 191,096.33 |
| 2_linear_2 | 150.3958 | 2019.7788 | 10,737.423 | 6940.350 | 723.1203 | 6940.350 | 197,200.00 |
| 2_linear_4 | 189.0303 | 2083.4211 | 10,163.768 | 8606.528 | 768.7449 | 8606.528 | 178,747.27 |
| 2_simple_2 | 157.3596 | 2042.6165 | 10,749.523 | 7308.142 | 762.7901 | 7308.142 | 196,300.00 |
| 2_simple_4 | 199.9375 | 2151.9671 | 9908.329 | 9159.259 | 798.1726 | 9159.259 | 165,511.25 |
| 3_exponential_2 | 81.1457 | 860.3691 | 2694.752 | 3578.629 | 370.9500 | 3578.629 | 31,575.38 |
| 3_exponential_4 | 89.3620 | 871.6876 | 2700.049 | 3635.239 | 380.0934 | 3635.239 | 30,548.94 |
| 3_linear_2 | 82.2700 | 871.9220 | 2717.038 | 3623.460 | 373.9550 | 3623.460 | 31,075.54 |
| 3_linear_4 | 92.8501 | 896.0751 | 2741.063 | 3683.842 | 380.8692 | 3683.842 | 29,327.73 |
| 3_simple_2 | 83.5511 | 888.1536 | 2752.627 | 3640.584 | 377.8576 | 3640.584 | 30,549.41 |
| 3_simple_4 | 98.7853 | 927.2893 | 2822.281 | 3703.337 | 417.6064 | 3703.337 | 28,478.29 |

Source: own elaboration.

The values presented in the tables show that in most cases, the lowest imputation error is generated by imputation method 3 with exponential weights and $k = 2$. (Method *3_exponential_2*).

Moreover, the results of the applied imputation methods were presented for two selected PPEs: Example4 and Example5. These energy consumption points have been selected to show the results of applying imputation methods for regular and irregular consumption. Figure 10 shows the actual consumption in April and May for PPE-Example4. It can be seen that the consumption is clearly cyclical with a cycle length of 1 week. The values on Saturdays and Sundays are clearly lower, while disturbances in the cycle at the beginning of May can be noticed.

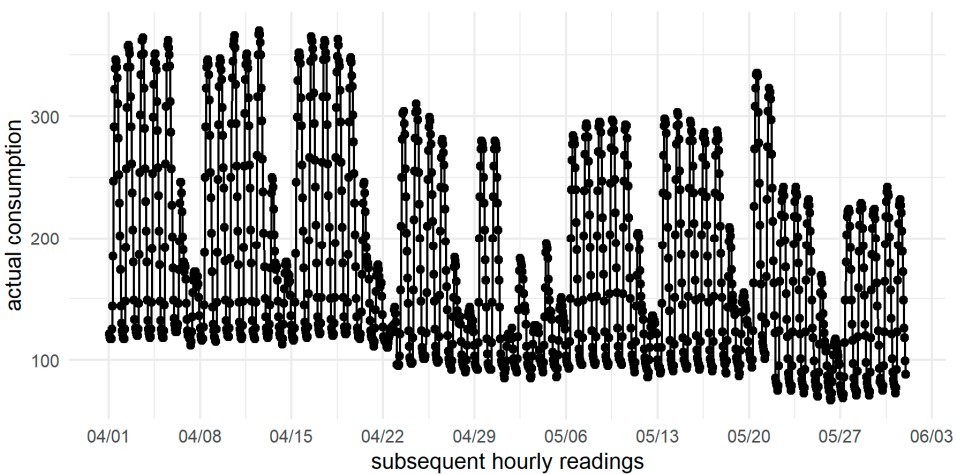

**Figure 10.** Actual consumption series for PPE-Example4. Source: own elaboration.

To show the performance of the imputation methods used, the missing values from 9 May to 12 May (Thursday–Sunday) were inserted in the 2019 series of data presented in Figure 10, as shown in Figure 11. The dotted line shows the locations of missing data.

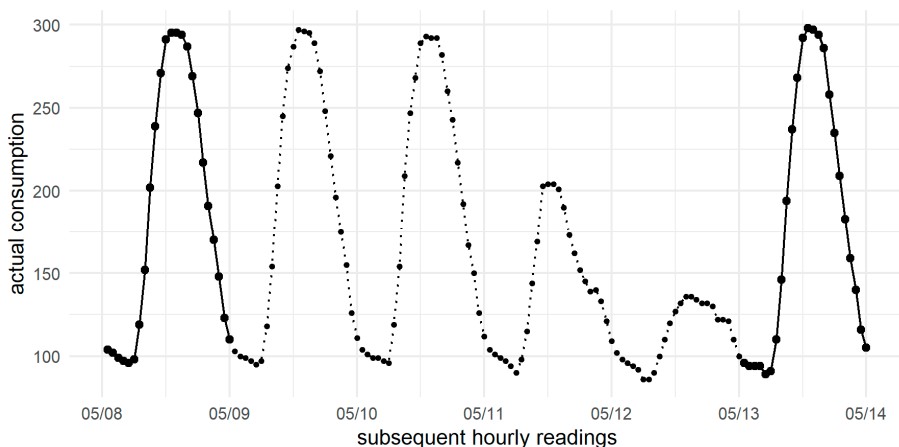

**Figure 11.** Actual consumption of PPE-Example4 with missing values inserted. Source: own elaboration.

Then the missing values were imported using the three imputation methods used, as shown in Figure 12. The gray line is actual consumption, the red line is imputation using method 1 (exactly), the yellow line is imputation method no. 2, the blue line is imputation method no. 3.

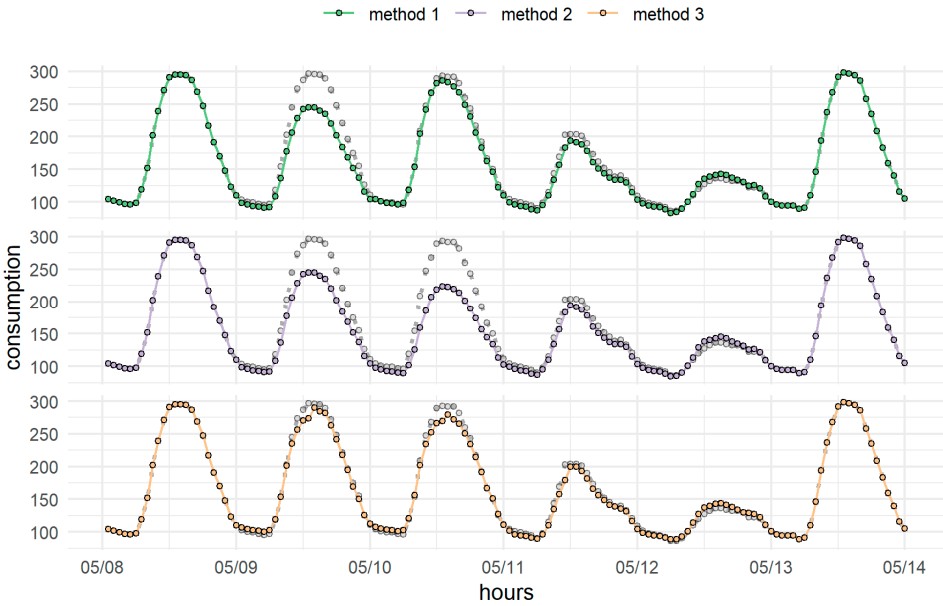

**Figure 12.** Actual consumption of PPE-Example4 and values after imputation according to the three imputation methods. Source: own elaboration.

As shown in Figure 12, method 1 and method 2 give the same results on 9 May and 11 May (Thursday and Saturday)—in the chart, the yellow line covers the red line. On Saturday and Sunday (11 May and 12 May), all three methods produce similar results. However, there is a clear difference on Thursday and Friday (9 May and 10 May). On Thursday (May 9), method 3 gives much better results than methods 1 and 2. However, on Friday, method 1 has a slight advantage over method 3, but both are clearly better than method 2.

To generalize the obtained results and determine the efficiency of the methods used in the presented example, the values of the mean imputation errors (MAE and MAPE) were calculated for each of the methods presented in Table 8. As shown in Table 8, the most efficient imputation method for the case of regular electricity consumption presented above (PPE-Example4) is method 3 (the MAE and MAPE error values are then the smallest).

**Table 8.** MAE and MAPE error values for individual methods.

| Method | MAE | MAPE |
|---|---|---|
| 1 | 10.8802 | 0.0570 |
| 2 | 18.6528 | 0.0927 |
| 3 | 5.8157 | 0.0342 |

Source: own elaboration.

The second example presented is irregular consumption for PPE-Example5. For this electricity consumption point, Figure 13 shows the actual consumption in April and May 2019. In this case (as shown in Figure 13), the consumption is irregular, and it is difficult to distinguish clear cycles.

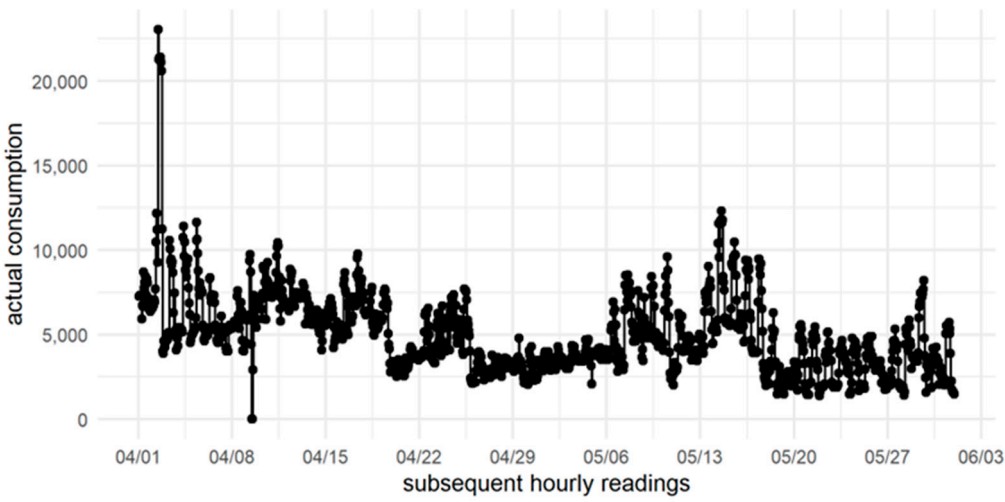

**Figure 13.** Actual consumption of PPE-Example5. Source: own elaboration.

As for the series with regular electricity consumption, missing data were inserted from 9 May to 12 May (Thursday–Sunday). The places of missing values are shown in Figure 14 (dotted line).

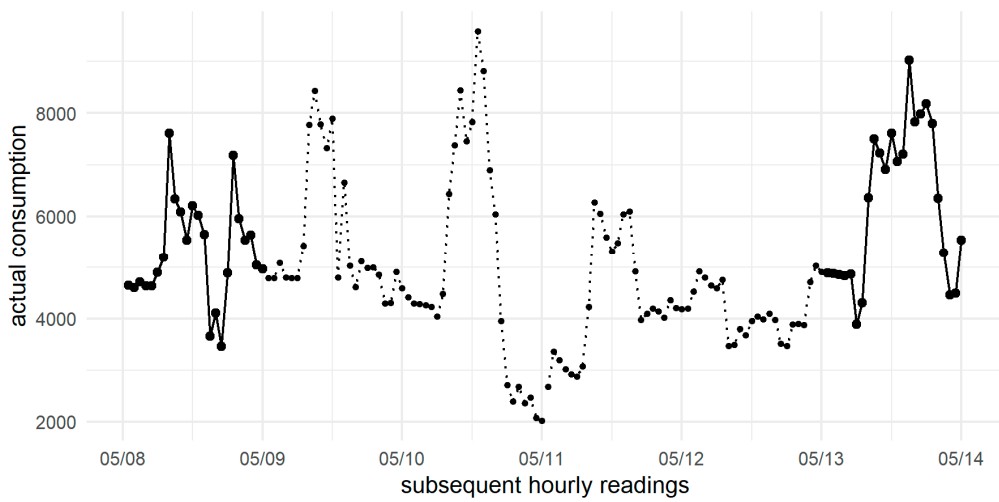

**Figure 14.** Actual consumption of PPE-Example5 with missing values inserted. Source: own elaboration.

The missing values were again supplemented with the use of the three analyzed imputation methods, as shown in Figure 15. The gray line is actual consumption, the red line is imputation method 1, the yellow line is imputation method no. 2, and the blue line is imputation method no. 3.

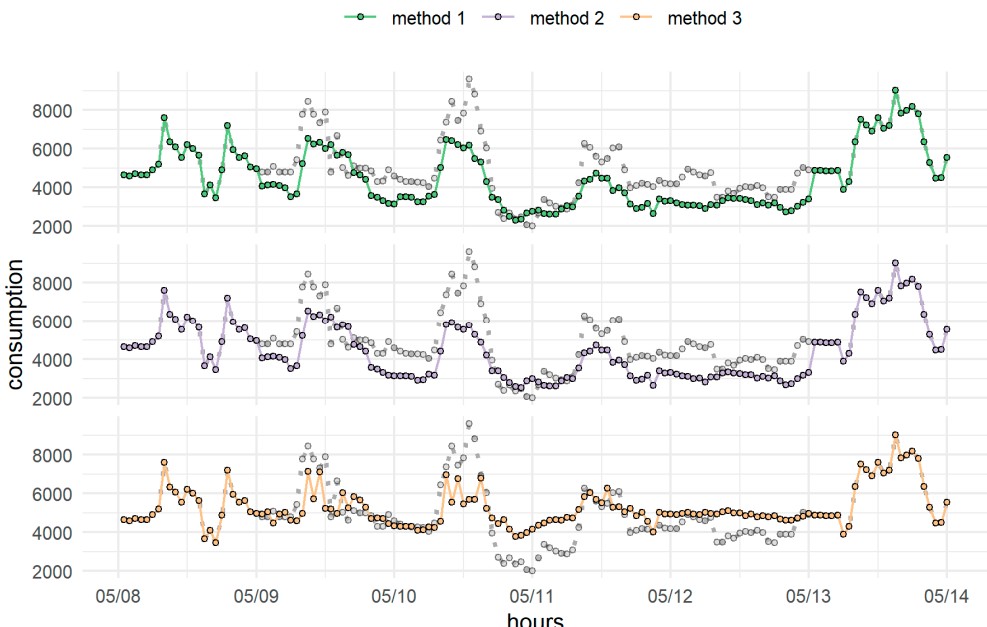

**Figure 15.** Actual consumption of PPE-Example5 and values after imputation according to three imputation methods. Source: own elaboration.

In the case of irregular consumption (Figure 15), it is difficult to clearly visually evaluate which method gives the best results. Mean imputation errors MAE and MAPE were again used to evaluate the methods. The values of these errors are presented in Table 9.

**Table 9.** MAE and MAPE error values for individual methods.

| Method | MAE | MAPE |
|:---:|:---:|:---:|
| 1 | 1062.1979 | 0.2130 |
| 2 | 1154.7951 | 0.2324 |
| 3 | 905.0303 | 0.2191 |

Source: own elaboration.

The smallest absolute error (MAE) was obtained for method 3, while in the case of relative error (MAPE), methods 1 and 3 give very similar error values, and for this particular analyzed data series (PPE-Example5), these values are better than for method 2.

In conclusion, we tested 3 methods of imputating data from the imputeTS package, with different variants, which resulted in 18 test cases. We did not test all the available algorithms in this package like Demirhan and Renwick [5] because we discovered seasonality in our time series. We also obtained different results from the previously mentioned authors for the hourly series, but the solar irradiance series analyzed by them did not show any seasonality, unlike the series of electricity consumption we analyzed.

The conducted analyses showed that the best performance in the case of univariate time series related to electricity consumption is provided by the imputation method with the use of seasonality decomposition with exponential weights and $k = 2$ (*method 3_exponential_2*).

## 5. Conclusions

There is extensive literature on electricity consumption data [21,22,43–46]. In most cases, the analysis of such data is aimed at more accurate forecasting of energy consumption, and thus at efficient and effective energy management. In this article, we also deal with the problem of electricity consumption, but in the context of trading companies that are responsible for ensuring a balance between the amount of energy purchased and the

amount of energy sold to customers. Too little or too much of the purchased energy, in relation to the sale, generates financial losses for the company. Moreover, efficient energy management is essential throughout the economy as it enables costs to be reduced and the activities of companies to grow sustainably.

Trading companies must determine the volume of electricity sales, and the basis in this respect is the sum of forecasts from concluded contracts. However, by applying methods based on historical data, it is not possible to improve the medium- or long-term demand forecast of the trading company in relation to the total electricity consumption by customers. The key, in this case, is the use of a method that improves the quality of individual forecasts for individual energy consumption points (PPE). The barrier faced by trading companies to increase the accuracy of forecasting for individual PPEs is missing values in the historical data. In this case, an estimate of the missing values in the historical time series has to be made and then the missing values should be replaced with these estimates, which is called missing data imputation or gap filling. As shown earlier, there are many methods and approaches for the imputation issue. However, it should be noted that univariate time series require an individual approach to data imputation problems as they do not contain additional attributes. We deal with this situation in the data analyzed by us, which does not contain additional information, such as, for example, in studies [17], where weather data was an additional attribute.

Therefore, we have proposed a procedure that allows to choose the appropriate imputation method in the analyzed case. First, we performed a statistical analysis of the occurrence of missing data and examined the distribution of missing data in terms of the moment of their occurrence. Then we chose three techniques for generating missing data Based on the analyses conducted, we also chose the methods and parameters of imputation. The data analysis carried out showed seasonality in the analyzed time series, therefore, we tested three methods of data imputation: the calendar method (Method 1), the imputation method by separating the phases of seasonal cycles (Method 2), and the imputation method using seasonal decomposition (Method 3). For each of the methods, we considered three ways to determine the weights: the exponential weighted moving average method, with the linear weighted moving average, the simple moving average, and two values of $k = 2$ and $k = 4$, which ultimately resulted in 18 variants of approaches to data imputation. The next step was to compare the selected methods and variants of imputation based on MAPE and MAE errors calculated for individual PPEs based on actual values and imputed values. The effect of using the proposed procedure is the selection of the best imputation method for the analyzed data. Detailed statistics of MAPE and MAE errors for the three methods of generating missing values and their variants indicated that in most cases, the lowest imputation error was generated by the third method using Seasonally Decomposed Missing Value Imputation with exponential weights and $k = 2$ (method 3_exponential_2). ImputeTS package was used because, as emphasized by Demirhan and Renwick [5], the use of this R packet is appropriate for one-dimensional data series. The mentioned authors analyzed the solar radiation intensity data, but their data, similar to the data analyzed in this article, had no additional attributes. In the analyzed hourly data of electricity consumption in this article, seasonality was detected, so not all methods of data imputation as in [5] were tested, but three methods that take into account seasonality. As mentioned earlier, hourly data were analyzed. Demirhan and Renwick did not detect seasonality for hourly data, hence other imputation methods are more effective in the case of hourly data than in the work of these authors.

Our research concerned data from trading companies and we hope that the conducted analysis will provide them with tools (methods) to deal with missing values, and thus contribute to the improvement of electricity consumption forecasts. In future research, the presented results will be used to work on the detection of anomalies in electricity consumption in relation to the forecasts. This will allow trading companies to better manage electricity orders in the long term and to monitor the electricity consumption of their customers on an ongoing basis. In this way, companies will be able to detect and

observe excessive jumps/drops in consumption, increases and decreases in consumption inconsistent with the forecasts, and correct them in such a way as to rationalize their own electricity orders on the exchange.

Increasing the credibility of forecasts may, on the one hand, contribute to a more precise balancing of electricity demand and production, and, on the other hand, may result in trading companies being able to offer consumers more favorable purchase prices for energy by minimizing part of the risk related to imbalance.

**Author Contributions:** Conceptualization, A.K.-S., J.S. and A.S.; Data curation, T.O.; Formal analysis, A.S. and M.W.; Investigation, T.O., A.S. and M.W.; Methodology, A.K.-S., T.O. and M.W.; Resources, J.S.; Visualization, T.O.; Writing – original draft, A.K.-S. All authors have read and agreed to the published version of the manuscript.

**Funding:** This work was supported by funds from a project called "Development of the Prototypical System of Electricity Consumption Anomaly Discovery Using Artificial Intelligence Tools to Streamline the Power Demand by Trading Companies", funded under the European Regional Development Fund, Subactivity 1.1.1: "Industrial Studies and Development Works Carried out by Companies" within the Operational Program Intelligent Development 2014–2020 (the competition held by the National Center for Research and Development). This research was also funded by Silesian University of Technology, grant number 13/010/BK_21/0057.

**Institutional Review Board Statement:** Not applicable.

**Informed Consent Statement:** Not applicable.

**Data Availability Statement:** Not applicable.

**Conflicts of Interest:** The authors declare no conflict of interest.

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
