# Peer review of "Analysis of Business Customers’ Energy Consumption Data Registered by Trading Companies in Poland"

_energies, doi:10.3390/en15145129_

Round 1
Reviewer 1 Report
The paper deals with an interesting topic that is in line with the key aims of the journal. The abstract is in line with the content of the paper: the main goal of the study, the methodology, and the results are described, however, the originality of the paper and the own contribution to the literature debate should be more emphasized. The structure of the article is easy to follow. The literature review is acceptable in general. The presentation of sampling, data, and methodology is fair. The methodology used by the authors fits the research questions. The discussion and conclusion session is general, and comparisons with previous literature results are neglected. It is recommended to clarify the limitation of the results. Quotation and referencing meet the formal requirements. In sum, the manuscript needs minor revisions.
Author Response
Dear Reviewer, thank you very much for your kind and constructive approach to the text we presented. Below is the detailed answer to your comment
Reviewer #1
The discussion and conclusion session is general, and comparisons with previous literature results are neglected. It is recommended to clarify the limitation of the results.
In this article, we deal with the problem of electricity consumption, but in the context of trading companies. As we have shown in Section 1 (introduction)), these data do not resemble the data widely discussed in the literature and do not have additional attributes. These are actual data possessed by trading companies. We did not find related works on the analysis of similar data, as obtaining data directly from commercial companies is rare. In the conclusions, we refer more broadly to the work of Demirhan and Renwick (2018) (changes are marked in yellow- lines 579-587). The data analysed by the above-mentioned authors concerned the intensity of solar radiation, but also had the form of one-dimensional series and resembled our data. We discussed also further research in the “Conclusion” section (changes are marked in yellow – lines 590-597). Moreover, in order to present the results better, we changed the layout of the sections. In the revised version of the article, we have the following main sections:
- Introduction
- Data
- Methods
- Results and discussion
- Conclusion
Reviewer 2 Report
"Analysis of business customers energy consumption data registered by trading companies in Poland" is an article providing a detailed discussion of the issues surrounding identifying an appropriate method of imputation missing data for this type of data. Trading companies expect a specific solution, so the authors proposed a procedure that allows choosing the imputation method, which will consequently improve the accuracy of forecasting energy consumption. The authors focus not only on theoretical aspects but also on applying the proposed approach in practice, making the article of interest to researchers and practitioners alike.
In my opinion, the background of the problem has been presented in an unambiguous way. The procedure of research is shown in the introduction. In my opinion, the article would benefit from the "2. Materials and methods" section, where the authors will describe in detail the research procedure and the data sources used.
Furthermore, the authors should not refer to numerous sources in one sentence, for example, in lines: 24, 25, 65/66. It looks like an artificial generation of numbers of positions in a bibliography. Each source should be discussed separately in the text.
It may also be of interest to briefly discuss further research in the conclusion section.
Author Response
Dear Reviewer, thank you very much for your commitment and thorough analysis of our research and for all your valuable comments. Below are the detailed responses to the individual issues.
Reviewer #2
Furthermore, the authors should not refer to numerous sources in one sentence, for example, in lines: 24, 25, 65/66. It looks like an artificial generation of numbers of positions in a bibliography. Each source should be discussed separately in the text.
This method of citation also appears in other articles from the Energies journal (this method is generally accepted in this journal). In addition, we also refer to many works cited in this way individually in the article.
Reviewer 3 Report
The article is correct in terms of content. The article analyzes data on energy consumption by business customers registered by trading companies in Poland. The area of ​​missing data in hourly series was correctly analyzed to determine the volume of electricity orders. The purpose of the article is given. The conclusions are correct. Relevant literature. The advantage is that the work was supported by funds from the project "Development of a prototype system for detecting an anomaly in electricity consumption using artificial intelligence tools to improve energy demand by commercial companies", financed under the European Regional Development Fund, Sub-measure 1.1.1: " Industrial research and development works carried out by companies ”under the Intelligent Development Operational Program 2014-2020 (competition of the National Center for Research and Development). These are the positive aspects of the article. Potential for improvement:
-) I suggest adding a short explanation of why it was decided to analyze in detail the data of business customers in Poland. Can we expect similar dependencies in other, e.g. neighboring countries? How is it in other countries?
-) Please explain why the data are from 2019. Currently (June, 2022), due to the situation, there is a high probability that the data look different.
-) Line 342-342. Is it a table or is it a drawing?
-) From my point of view, it is worth considering converting this article to an impersonal form.
-) I don't feel qualified to judge about the English language and style.
Overall, I recommend that you publish the article after making these small improvements.
Author Response
Dear Reviewer, thank you very much for your positive opinion and kind approach to the article we presented. Below are the detailed responses to the individual issues.
Reviewer #3
Line 342-342. Is it a table or is it a drawing?
This is not a table or a drawing. This is just a step-by-step description of the procedure shown in Figure 5. We have corrected it in the text.
Reviewer 4 Report
Dear Authors,
This is a good article, but it needs some improvements:
1. What are the hypothesis in the article?
2. In which way the statistics results have impact on practice?
3. What are the policy implications from the research?
4. Please, prepare monenclature of abbreviations used in the paper.
5. Please, write sources under the tables and figures.
Author Response
Dear Reviewer, thank you very much for your commitment and thorough analysis of our research and for all your valuable comments. Below are the detailed responses to the individual issues.
Reviewer #4
Please, prepare nomenclature of abbreviations used in the paper.
Dear Reviewer, we believe that it is not necessary to prepare a nomenclature of abbreviations, as all abbreviations have been carefully explained in the text, especially in the "Methods" section. Moreover, in order to present the results better, we changed the layout of the sections. In the revised version of the article, we have the following main sections:
- Introduction
- Data
- Methods
- Results and discussion
- Conclusion
Reviewer #5
Please, write sources under the tables and figures.
We thank you for this comment. Sources under tables and figures have been added